# Is There a Universal Endurance Microbiota?

**DOI:** 10.3390/microorganisms10112213

**Published:** 2022-11-09

**Authors:** Hope Olbricht, Kaitlyn Twadell, Brody Sandel, Craig Stephens, Justen B. Whittall

**Affiliations:** Department of Biology, Santa Clara University, Santa Clara, CA 95053, USA

**Keywords:** 16S gene sequences, barcode, network, *Romboutsia*, *Veillonella*, workflow

## Abstract

Billions of microbes sculpt the gut ecosystem, affecting physiology. Since endurance athletes’ performance is often physiology-limited, understanding the composition and interactions within athletes’ gut microbiota could improve performance. Individual studies describe differences in the relative abundance of bacterial taxa in endurance athletes, suggesting the existence of an “endurance microbiota”, yet the taxa identified are mostly non-overlapping. To narrow down the source of this variation, we created a bioinformatics workflow and reanalyzed fecal microbiota from four 16S rRNA gene sequence datasets associated with endurance athletes and controls, examining diversity, relative abundance, correlations, and association networks. There were no significant differences in alpha diversity among all datasets and only one out of four datasets showed a significant overall difference in bacterial community abundance. When bacteria were examined individually, there were no genera with significantly different relative abundance in all four datasets. Two genera were significantly different in two datasets (*Veillonella* and *Romboutsia*). No changes in correlated abundances were consistent across datasets. A power analysis using the variance in relative abundance detected in each dataset indicated that much larger sample sizes will be necessary to detect a modest difference in relative abundance especially given the multitude of covariates. Our analysis confirms several challenges when comparing microbiota in general, and indicates that microbes consistently or universally associated with human endurance remain elusive.

## 1. Introduction

Endurance has played an important role in human history from our origins in the African savannah, through historical times (i.e., the origin of the marathon) to the more recent fascination with ultra-endurance events. Success in endurance events is assumed to be a product of training, genetics, and psychological preparation to withstand extreme mental and physical challenges [1]. Physiologically, the factors limiting performance in endurance events have traditionally been divided into two broad categories—aerobic (i.e., VO_2_ max) and anaerobic (i.e., lactic acid) [2]. However, what if one of the limiting factors to success in endurance events was not human at all?

We are as microbial as we are human. Bacterial cells associated with the human body are at least equivalent in number to human cells, if not more abundant [3]. More than 1000 bacterial species may be found in the intestines of each person, and over 70% of the total human microbiome is contained in the gut [4]. Whereas human cells are slowly replaced with identical (or nearly identical) copies, the cells of the gut microbiota are constantly changing due to immigration, emigration and differential rates of division based on the dynamic gut environment, to which they are much more sensitive and responsive than human epithelial cells. If our fuel tanks are lined with bacteria, it is not surprising that gut microbiota may affect performance in fuel-limited sports such as endurance events. Furthermore, the dichotomy between human physiology and microbial metabolism is increasingly difficult to disentangle, as more human genotype/microbiome interactions continue to be discovered [5].

In some studies, exercise has been associated with increased gut microbial diversity, increased *Bacteroidetes*–*Firmicutes* ratio, and proliferation of bacteria which can modulate mucosal immunity and improve barrier functions [6,7,8,9,10,11]. All of these changes could contribute to increased performance, decreased inflammation, decreased gastrointestinal (GI) distress, and faster recovery times [6], as well as possible protection against GI infections [7,8,9,10,11]. However, factors other than intensive exercise could also contribute to these reported changes in the microbiota of endurance athletes. In fact, numerous factors have been suggested to affect the composition of the gut microbiota, including (but not limited to) method of birth, diet, sex, age, antibiotic use, geographic region, stress level, and disease history [12]. For example, dietary behaviors such as carbohydrate loading, a traditional prelude to an endurance event, could increase *Prevotella* independent of physical exertion [13]. Consumption of probiotics could also affect the microbiota irrespective of exercise level (for example, see [14]). A recent review found mixed effects of probiotics on performance, as 17 studies showed no significant results of probiotic consumption and seven showed an improvement in performance (summarized in [13]). Probiotics may also protect against some upper respiratory tract infections which in athletes could result in longer stretches of continuous, intensive training [15]. Interactions among bacteria may be important, as multi-strain probiotics are more likely than single strains to lead to improved endurance performance [13]. The mixed results regarding the effects of probiotics on athletic performance indicate that there may be an interaction between gut bacteria and endurance, but warrant further study.

The concept of an “endurance microbiome” suggests there are assemblages of gut bacteria that are more common in athletes compared to sedentary controls, or gut bacteria that are enriched after an endurance event compared to before the event. The hypothetical endurance microbiota could be characterized by changes at multiple levels such as (1) increased overall diversity, (2) different levels of abundance, and (3) the emergence of beneficial bacterial associations. Below, we briefly summarize these three metrics.

To date, studies on the effects of exercise on gut microbiota diversity have exhibited mixed results. Some investigations describe increased microbial diversity associated with exercise [6,16,17,18,19,20], while others report no significant change in diversity [21,22,23,24]. Lensu and Pekkala [25] have conducted a recent and thorough literature review of the effect of exercise on the gut microbiota concluding that exercise has a “beneficial effect on the gut microbiota” and is associated with “healthy gut microbiota”. However, when the relative abundance of individual bacterial genera is examined, there is very little consistency among the studies they reviewed. Deciphering these inconsistent results requires homogenizing the bioinformatics pipeline and subsequent statistical analyses.

Therefore, we reanalyzed the raw data from three studies (four comparison groups) that all utilize 16S barcoding data collected on the Illumina platform (at least in part) yet have come to largely different conclusions regarding the changes in relative abundance associated with endurance exercise. First, in a comparison of runners before and after the Boston Marathon, Scheiman et al. [26] found only *Veillonella* exhibited a significant increase in abundance after the marathon. The authors then showed that *Veillonella* improves endurance performance by metabolizing lactate that has crossed the blood–gut barrier into short-chain fatty acids that can improve muscle performance (in mice) [26]. Second, when Zhao et al. [23] examined Chinese half-marathon runners, they found 12 genera whose abundance increased after the event, not including *Veillonella*. Third, Peterson et al. [18] compared competitive cyclists with variable training intensities and found some of the same genera as Zhao et al. [23], yet identified several new genera with differential abundance based on 16S and transcriptome data. These studies identified a diversity of candidate genera that may comprise endurance-associated microbiota, but due to methodological differences in their bioinformatic workflows, it is difficult to make direct comparisons among the studies.

Endurance-associated microbiota could also manifest in associations among bacteria, or groups of bacteria. For example, several studies have highlighted a putative trade-off between two common groups of gut bacteria implicated in endurance athletes—*Prevotella* and *Bacteroides* [27,28]. Other investigations have found strong correlations among bacterial lineages and in response to environmental stresses [29]. Many exercise-related studies have looked for inter-bacterial associations. *Prevotella* has been implicated in athletic performance and associated with *Streptococcus*, *Enterococcus*, *Desulfovibrio*, *Lachnospiraceae*, *Succinivibrio*, *Oscillospira*, *Xylanibacter*, and *Butyrivibrio* [20]. Nevertheless, Gorvitovskaia et al. [28] report no consistent bacterial correlations among the four studies they reviewed. Potentially, bacterial interactions associated with endurance are broader than pairwise correlations, perhaps represented better as a connectivity network of the entire microbiota community. Active people who exercise regularly have been reported to have more complex gut bacterial networks than sedentary controls [17]; however, the connectivity networks underlying endurance-associated microbiota are largely unexplored [30], and no meta-analysis has compared networks using the same methodology across datasets.

We hypothesize that if a “universal” endurance-associated assemblage of microbes exists, it should manifest regardless of geographic location, type of sport, or specific endurance event. Our goal is to reanalyze several relevant studies using a single bioinformatics pipeline and consistent downstream statistical analyses to determine whether there are repeated changes in diversity, relative abundance, or associations between genera in response to intensive endurance exercise. The variance in bacterial abundances were used in a power analysis to determine the necessary sample sizes to detect a modest difference. 

## 2. Materials and Methods

### 2.1. Datasets

After searching the literature for relevant studies (Appendix A), we selected and reanalyzed four gut microbiota datasets involving endurance athletes from three previously published studies (Table 1). All studies utilized Illumina sequencing of the V3-V4 region of the 16S rRNA gene (amplicon sequencing) [18,23,26]. Below is a brief description of each dataset.

#### 2.1.1. Boston Marathon Study

Scheiman et al. [26] recruited 15 elite athletes running in the 2015 Boston Marathon, along with 10 sedentary controls. They conducted amplicon sequencing on 209 fecal samples taken daily from participants up to one week before to one week after the marathon using Earth Microbiome Project primers targeting the v4 region of the 16S rRNA gene [using primers 515F (Caporaso) and 806R (Caporaso); https://earthmicrobiome.org/protocols-and-standards/16s/ (accessed on 4 July 2022)]. Amplicons of approximately 292 bp (based on *E. coli* 16S rRNA REFSEQ NR_024570) were sequenced using 150 bp Illumina paired-end reads and were processed with the DADA2 pipeline and phyloseq. Generalized linear mixed-effect models and leave-one-out cross validation were used to determine significant associations. According to their supplemental data, some samples were “rerun” on the Illumina sequencer. In these cases, we only used the data from the reruns (i.e., SG10 and SG27).

#### 2.1.2. Chongqing Half-Marathon Study

Zhao and colleagues [23] recruited 20 amateur athletes who were running in the 2016 Chongqing International Half Marathon. A total of 40 fecal samples were collected—each runner was sampled the morning before the race and again after the race. Zhao et al. used very similar reverse primers to Scheiman et al.; however, they added CC to the 3′ end and used T instead of the ambiguity W nine base pairs from the 3′ end. Their forward primer lands 176 bp further upstream than the Scheiman et al. primers generating a ~465 bp amplicon (based on *E. coli* 16S rRNA REFSEQ NR_024570). Because of the larger amplicon length, they collected 250 bp paired-end reads generated on an Illumina HiSeq. Two samples from runner nine were eliminated from our reanalysis because of ambiguous labeling of the data. Each participant was given the same kind of food during the period between the first and second sample collection.

#### 2.1.3. Competitive Cyclist Study

Petersen et al. [18] studied 33 competitive cyclists categorized into four non-overlapping training groups based on their average training time per week: 6–10 h, 11–15 h, 16–20 h and 20+ h per week. A total of 33 samples were collected, one from each cyclist. They collected both whole-genome shotgun sequence data and 16S rRNA gene amplicon sequence data. We reanalyzed the latter to compare to the two studies (Petersen et al. 2017, Additional File 1). Petersen et al. used 150 bp paired-end Illumina reads from 16S rRNA amplicons of the v4 hypervariable region with primers comparable to Scheiman et al. [24]. We grouped the cyclists into low (6–10 h/wk; *n* = 8), medium (11–15 h/wk; *n* = 17), and high (16–20+ h/wk; *n* = 8) categories for diversity analyses, then focused on the two most extreme training groups (with balanced sample sizes) when searching for a universal endurance microbiota (e.g., “low” vs. “high” training). 

#### 2.1.4. Sample Collection, Storage and DNA Extraction

Fecal sample collection and storage and affect the estimates of relative abundance from 16S amplicon sequencing [31]. Fecal samples from all three studies reanalyzed herein were self-collected; however, Scheiman et al. [24] and Zhao et al. [23] used polypropylene tubes for collection and stored samples at 4 C short-term, while Petersen et al. [18] used polyethylene tubes and stored samples with “frozen freezer packs” short-term. All three studies −80 °C for long-term sample storage. Although different DNA extraction methods were used across all three studies, Rintala et al. demonstrate that the impact of different DNA extraction methods on relative abundance estimates from 16S amplicon sequencing is relatively minor [31].

### 2.2. Target Genera

Among the Boston Marathon, Chongqing half marathon, and competitive cyclist studies, there were 1, 12, and 6 bacterial genera identified as having significantly different abundances between treatment groups, respectively (Appendix A). Of these, no single genus was identified in all three studies and only three genera were found in two of these studies. We used the 16 unique genera from all three studies as our “target genera” in a hypothesis-testing framework (using alpha < 0.05 as a cutoff). Subsequently, we expanded our analyses to all remaining genera, and corrected for multiple tests, since the comparisons were not based on a priori hypotheses (i.e., using the Benjamini–Hochberg false-discovery-rate correction—see below for details). 

The target “genera” from Zhao et al. [23] included both individual species (e.g., *Prevotella corporis* which we treated as *Prevotella*) and genera. The authors emphasized the significantly differential abundance in the family Coriobacteriaceae before and after the half marathon, but we only included *Collinsella* since we assume this genus was driving the significant result based on their Figure 2B. We did not include “unclassified Porphyromonadaceae’’. Finally, they report “*Phaseolus vulgaris*” in their Figure 2 results [23], but they discuss *Romboutsia* later in their findings. We assume this is a technical mistake (*Phasaeolus vulgaris* is a species of legume, not a bacteria). *Phaseolus* sensu Zhao et al. [23] is hereafter treated as *Romboutsia*.

### 2.3. Microbiome Assembly (Bioinformatics Pipeline)

Although there are many published tools for measuring bacterial abundance from 16S rRNA gene amplicon sequencing using the Illumina platform (e.g., QIIME [32]; MOTHUR [33]; DADA2 [34]), we developed a simple workflow in Geneious Prime 2021.2.2 by adapting their Amplicon Metagenomics tutorial [(https://www.geneious.com/tutorials/metagenomic-analysis/ (accessed on 1 May 2020)]. Raw data from published studies was downloaded from usegalaxy.org in the form of fastq files and imported into Geneious Prime. We used Illumina paired-end, inward pointing reads with sequences interlaced within each fastq file. The minimum quality (q) cutoff was empirically determined to be 13 in a pilot study, and the merge rate was set at “very high” to maximize reads mapped, while reducing incorrectly mapped reads. After using BLAST to Genbank (Release 242) for OTU clustering to construct a reduced 16S rRNA sequence database per dataset, we mapped reads back onto the reduced database to classify the reads to genus based on 90% minimum overlap identity per Geneious’ Amplicon Metagenomics tutorial and their Sequence Classifier tutorial. Although 95% 16S rRNA gene sequence identity has often been used as a cut-off for bacterial genus-level operational taxonomic units [35,36,37,38], our more liberal cut-off was intended to maximize the number of classified reads while accounting for the sequence variation within genera. Using this cutoff, we were not trying to detect bacterial species, only genera for consistency in making comparisons across studies. All subsequent analyses were based on the relative abundance which is the proportion of reads mapped to a genus compared to total reads mapped per sample (per Gloor et al. [39]). Classifications were pruned to genus, with all higher taxonomic-level BLAST results removed. Our Geneious workflow is available from FigShare (DOI: 10.6084/m9.figshare.c.6036347). 

### 2.4. Diversity

We measured the alpha diversity for each dataset and each treatment group as the number of unique genera identified. We then used Simpson and Shannon indices to compare diversity considering each genus’ relative abundance in the vegan package (v. 2.6-2) in R Studio (2022.02.3). For the competitive-cyclist study [18], diversity measures among the three independent treatment groups were compared using an ANOVA in R. All other diversity comparisons for the Boston marathon and Chongqing half-marathoners were carried out using *t*-tests on independent samples (athletes vs. controls) or paired samples (before vs. after) in R. 

### 2.5. Relative Abundance Comparisons

#### 2.5.1. Normality Testing

The relative abundance of each bacterial genus in each dataset was assessed for normality using the Shapiro–Wilk test. Mean kurtosis and mean skewness, two descriptors of the relative abundance distributions, were also recorded. After finding most bacterial genera had relative abundance distributions significantly different from normal and were outside the recommend range of normal kurtosis and skewness values (−2 to +2), the relative abundances were square-root-transformed and reassessed for normality, kurtosis and skewness. Again, most genera failed to meet the assumptions of normality, so we used non-parametric statistical tests of the untransformed relative abundance data henceforth.

#### 2.5.2. Overall Microbiota Community Comparisons

Differences in the relative abundances of the entire bacterial communities within and between treatment groups were evaluated using Euclidean distances in ANOSIM for each of the four datasets. To determine if the bacterial communities were significantly different between treatment groups (overall), we performed 9999 permutations. Distances within and between microbiota communities were visualized using MDS with the maximum number of tries set to 500 to improve convergence. Goodness-of-fit was determined based on the stress values (all have stress << 0.15 indicting a “good fit” or reasonable representation of the data). ANOSIM and MDS were conducted using the vegan package (v. 2.6-2) in R.

#### 2.5.3. Hypothesis-Driven Approach

For each dataset, relative abundances for target genera were compared using the Wilcoxon test with continuity correction in R (alpha = 0.05). Since the Boston marathon study [26] had multiple samples collected before and after the event, the average relative abundance before and after was used, as well as averages for the multiple samples taken from the sedentary controls. By using relative abundance, we standardize for sequencing depth which varies per sample (i.e., accounting for the compositional nature of the data). Although analyzing changes in relative abundance (“deltas”) corrects for differences in magnitude, thereby facilitating comparisons across taxa with different abundances, to statistically assess changes in relative abundance, we used the relative abundance data directly to avoid the additional data transformation step.

#### 2.5.4. Data Exploration

To explore previously unreported differences in bacterial abundance among these datasets, relative abundances between treatment groups were examined for all remaining genera detected in the gut microbiota (implemented in R). We applied the Benjamini–Hochberg for controlling the false-discovery rate since the Bonferroni correction is overly conservative (missing significant correlations when they exist or elevated Type 1 error) especially when a large number of tests are conducted and the hypotheses are highly correlated (e.g., microbiota community relative abundance data). The more powerful Benjamini–Hochberg method provides an appropriate balance between discovery of statistically significant results while limiting false positives (Type 2 error). Boston marathon control vs. athletes = 282 tests; Boston marathon “athletes before” vs. “athletes after” = 282 tests; Chongqing half marathon = 198 tests; and competitive cyclists = 148 tests.

### 2.6. Changes in Bacterial Associations

#### 2.6.1. Correlated Abundances and Changes Therein

Correlations in relative abundance between bacterial genera were tested for each treatment group individually. The analyses were restricted to the genera detected in at least 75% of all samples to avoid spurious correlations caused by large amounts of missing data (i.e., genera with no reads mapped). Spearman’s rank correlation coefficients, *p*-values and Benjamini–Hochberg false-discovery-rate correction were calculated in R Studio (2022.07.2). Boston marathon “athletes before”, “athletes after” and controls = 1596 tests; Chongqing half marathon “athletes before” and “athletes after” = 3081 tests; and competitive cyclists “low”, “medium” and “high” = 861 tests.

#### 2.6.2. Networks of Bacterial Associations (NetCoMi)

Potential microbe–microbe association networks underlying the gut microbiota of endurance athletes were explored using NetCoMi [30]. We looked for predictable changes in the connectivity of microbial association networks among treatment groups using quantitative comparisons across networks. Differences in the hierarchical clustering of bacterial interactions were examined between controls vs. athletes, athletes before vs. after and cyclists involved in differing levels of training. Raw read counts from the Geneious bioinformatics pipeline were pruned to genera, then filtered for samples with 1000 reads or more and limited to the 100 genera with the highest number of reads [30]. We constructed the network with SparCC [40] which accounts for the compositional nature of the data by applying a centered log-ratio transformation (clr) [41,42]. Furthermore, SparCC is less likely to identify spurious correlations compared to Pearson correlations [43]. Singular nodes were removed when present in one network and when comparing multiple networks. 

A *t*-test was applied (alpha = 0.001 for Boston marathon samples [26]; alpha = 0.1 for Chongqing half-marathon samples [23] and for the cycling study [18]) to reduce the network to a tractable size with a local false discovery adjustment [44] to select which edges to include in the network (“sparsification” [42]). To analyze the network, we applied the fast greedy modularity optimization algorithm for finding community structure (“cluster_fast_greedy” [45]) with hubs defined based on their eigenvalue (degree of connectedness to other nodes with high connectedness). A high eigenvector of centrality score means that a node is connected to many nodes who themselves have high eigenvector scores [46].

For each treatment group individually, we compared the relative size of the largest connected component of the resulting network (LCC = the connected component with highest number of nodes), betweenness centrality (the degree to which a node lies on paths between other nodes), closeness centrality (distance between a node and all other nodes), and degree centrality (# of edges = measure of co-occurrence) [47]. We also assessed dissimilarity (1—edge weight) and average path length for each of the three treatment groups.

When comparing networks, the layout represents the optimal “union” of each pair of networks (athletes before vs. after; controls vs. athletes after; and low- vs. high-training groups). Node size represents the centered log-ratio transformation of the number of reads per genus and node color distinguishes different clusters.

To compare two networks at a time (e.g., “athletes before vs. after” and “controls vs. athletes”), we examined network size (LCC), positive edge percentage (positively biased), hub taxa, adjusted Rand index [41], and the top 10 taxa showing the largest differences in three centrality measures (degree, betweenness and closeness). 

We conducted several statistical comparisons for these pairs of networks (“athletes before” vs. “athletes after” and controls vs. “athletes after”) using CompareNet in NetCoMi [30]. To test whether the two similarity matrices were significantly different from one another, we examined Jaccard indices for degree, betweenness, closeness, eigenvector, and hub taxa. A Jaccard index equal to zero indicates completely different matrices. In addition, we used the adjusted Rand index (ARI) which measures the similarity between clusterings (ARI = 0 means random clusterings in the two networks being compared; ARI = 1 means perfect agreement between clusterings in the two networks being compared) [48]. For Rand index, a *p*-value below 0.05 means that ARI is significantly higher than expected for two random clusterings based on 1000 permutations. We also used permutation tests (*n* = 1000) to determine if any genera showed significant differences in their degree, betweenness and closeness when comparing networks. Briefly, to generate a null distribution, the treatment group labels were randomly reassigned to the samples keeping the group sizes constant. The network metrics are then re-estimated for each permutation. We applied the local false-discovery-rate correction (lfdr) to correct for multiple testing.

We identified differences between the networks using Diffnet in NetCoMi [30]. To assess significantly different associations in the network, we applied Fisher’s Z-test after adjusting the *p*-value using the local false-discovery rate. Occasionally, this produced an empty network (e.g., Scheiman “athletes before” vs. “athletes after”), in which case, we loosened the filter in NetConstruct to ensure enough overlapping bacterial genera were able to be compared in the Diffnet analysis. 

R scripts for all data analyses are available from FigShare (DOI: 10.6084/m9.figshare.c.6036347).

## 3. Results

The bioinformatics pipeline assembled and mapped most of the raw reads. For the Boston marathon and competitive cyclist samples, ~60% of reads mapped to a bacterial genus. The remaining 40% did not meet the 90% identity threshold for known genera and were removed. For the half-marathoners, ~80% of reads mapped, reflecting the longer read lengths and thus increased likelihood of accurately mapping to a single genus.

### 3.1. Diversity

Overall diversity was measured as the total number of genera detected (richness) and Simpson and Shannon’s diversity indices. Variation in the number of genera detected was much higher among datasets than between treatment groups within each dataset (Table 1). We found 280 genera among the three Boston marathon treatment groups, 197 genera between the two Chongqing half-marathon treatment groups, and 148 genera among the three cyclist training groups, similar to what these studies reported previously (143, 317 and 200, respectively) (Scheiman et al. [26] Appendix A; Zhao et al. [23]; Petersen et al. [18] Additional File 1). In all datasets, a small number of genera make up the bulk of the microbiota (Figure 1). Several genera had high abundance in all datasets. For example, *Faecalibacterium* was consistently one of three to five genera that together comprised greater than 50% of the microbiota in treatment groups across datasets. In addition, *Bacteroides* and *Blautia* are very abundant in most treatment groups. However, there were also numerous differences. For example, *Prevotella* exhibited a very large increase in relative abundance in the medium and high cyclist training groups (Figure 1G,H) in comparison to the low-training group (Figure 1F).

To statistically compare overall diversity, we tested for differences between treatment groups using Simpson’s and Shannon’s indices (Appendix A). Among the four treatment group comparisons, none were significantly different for neither Simpson’s nor Shannon’s indices (*p* > 0.05).

### 3.2. Relative Abundance

#### 3.2.1. Normality Testing

Relative abundance data failed Shapiro–Wilk tests for normality in all but two genera among the three previously published datasets (Shapiro–Wilk Test, *p* < 0.05; Appendix A). For the untransformed data, all datasets had mean kurtosis and skewness were outside the recommended bounds for normality (−2 < x < +2). Following square root transformation, ~25x more tests passed the Shapiro–Wilk test for normality, but still less than 50% of the genera tested. Following square root transformation, skewness data from all three studies were within the bounds of normality, but mean kurtosis was within the recommended range for normality for only one out of three study datasets. Since the square-root-transformed data still failed to meet the assumptions of normality for most genera in all three study datasets, we chose non-parametric tests when performing statistics with relative abundance data.

#### 3.2.2. Overall Microbiota Community Comparisons

When comparing the overall microbiota communities between treatment groups using non-parametric statistics, only one of the four datasets detected a significant difference (cyclists low- vs. high-training groups) (Table 2; ANOSIM, r-value = 0.5173, *p* = 0.0012). Multidimensional scaling (MDS) clearly shows substantial overlap among the samples of the two treatment groups in three out of the four datasets (Figure 2). MDS stress values < 0.15 indicate a good fit of the data (Table 2).

#### 3.2.3. Hypothesis-Driven Approach

Prior results from the three datasets identified 16 bacterial genera with significant differences in relative abundance in the endurance-associated microbiota (Appendix A). When we examined the relative abundance for these 16 target genera using our bioinformatics pipeline, some were not even detected in all four datasets. For example, *Ezakiella* was not identified in any of the datasets (no reads mapped; Appendix A), although it was confirmed to be present in the BLAST database used to classify sequences in the Geneious workflow. *Ruminiclostridium* and *Actinobacillus* were missing from the Chongqing half marathon and competitive cyclist datasets. *Methanobrevibacter* and *Pseudobutyrivibrio* were not found in the cyclist dataset and *Akkermansia* was not detected in the Boston marathon dataset.

Among the ten target genera detected in all three datasets, none exhibited significant differential abundance in all four treatment group comparisons (Table 3). Only two genera had significantly different relative abundances in more than one of the treatment group comparisons (*Romboutsia* in the half-marathoners and cyclists, and *Veillonella* in the half-marathoners and marathoners control vs. “athletes after” comparison) (Figure 3). The following sections report results for each dataset individually.

In the Boston marathon dataset, there were two treatment group comparisons, and each had one target genus (out of the fourteen target genera detected) that exhibited significant differences in relative abundance (Figure 3, Appendix A, Table 3). In comparing “athletes after” to sedentary controls, the average proportion of *Veillonella* in “athletes after” is 22.5-fold larger than in controls (Figure 3, Table 3 and Appendix A, Wilcoxon Test, W = 129, *p* = 0.002). When comparing “athletes before” vs. “athletes after”, the average proportion of *Clostridium* was 3.6-fold larger in “athletes after” than in “athletes before” (Appendix A, Table 3 and Appendix A, Wilcoxon Test, V = 24, *p* = 0.041).

In the Chongqing half-marathon dataset, eight of the thirteen target genera detected showed significantly different abundances when comparing before and after the half marathon (Figure 3 and Appendix A, Table 3). The four most significant results stand out from the rest (Appendix A). Three of these four genera showed increased abundance after the event (*Coprococcus*, *Veillonella*, and *Collinsella*) ranging from 1.97 to 2.67-fold higher (Figure 3 and Appendix A). In particular, *Veillonella* had 2.67-fold higher abundance after the event compared to before the event (Table 3 and Appendix A; Paired Wilcoxon Test, V = 14, *p* = 0.0004). The fourth significantly different genus, *Romboutsia,* decreased approximately two-fold after the event (Figure 3, Table 3 and Appendix A; Paired Wilcoxon Test, V = 179, *p* = 0.0002).

Comparing athletes between the low (*n* = 8) vs. high (*n* = 8) training groups from the competitive cyclist dataset, we found three target genera with significantly different relative abundances (Figure 3 and Appendix A, Table 3). *Prevotella* had 700-fold higher abundance in the high-training group compared to the low-training group (Figure 3, Table 3 and Appendix A; Wilcoxon test, W = 1, *p* = 0.0003) and *Romboutsia* had approximately 8-fold higher abundance in the high-training group compared to the low-training group (Figure 3, Table 3 and Appendix A; Wilcoxon test, W = 6, *p* = 0.005). In contrast, *Bacteroides* showed a 39% decreased abundance in the high-training group compared to the low-training group (Table 3 and Appendix A; Wilcoxon test, W = 57, *p* = 0.007).

#### 3.2.4. Data Exploration (All Pairwise Comparisons)

After expanding our analyses to compare all genera detected in all four treatment group comparisons, only one genus had significantly different relative abundance after Benjamini–Hochberg correction (Appendix A). *Romboutsia* (one of our 16 target genera) from the Chongqing half-marathon dataset (see Hypothesis-Driven Approach section above) was ~2x lower after the event compared to before the event (Table 3).

### 3.3. Changes in Bacterial Associations

To determine if potential endurance-associated microbiota have distinct associations among bacteria, we first tested for all pairwise correlations in relative abundances between bacterial genera detected across datasets. Second, we used hierarchical clustering to identify networks of bacterial associations to determine whether there are any significant differences among treatment groups.

#### 3.3.1. Correlated Abundances

Among the sixteen target genera, there were only three significant correlations in relative abundance across all treatment group comparisons (Figure 4). *Coprococcus-Ruminococcus* was positively correlated in the Boston Marathon Control dataset, whereas significant negative correlations were detected between *Clostridium-Eubacterium* and *Prevotella-Bacteroides* in the half-marathon dataset. There were no consistently significant correlations emerging (or disappearing) in the endurance treatment group across these datasets (Figure 4).

Although we did not detect any consistent, significant endurance-associated changes in pairwise bacterial associations across the datasets, we were able to reconstruct the negative correlation between relative abundance of *Prevotella* and *Bacteroides* (Figure 4B,D,F,H) previously reported by numerous authors [10,16,18,20,28,49,50]. The negative correlation between these two cosmopolitan genera provides an internal control for our methodology since it has been reported in a range of studies outside of endurance athletes e.g., [20].

After expanding the Spearman’s rank correlations to all pairwise comparisons, only the half-marathon dataset showed significant correlations emerging after the event. There were 28 significant correlations after the half marathon that were not correlated before the half marathon (Appendix A).

#### 3.3.2. Networks of Bacterial Associations (NetCoMi)

The bacterial association networks are statistically similar between treatment groups within a dataset and very different across datasets (Appendix A). First, comparing treatment groups within each dataset, the Rand index (which measures the similarity of two networks by randomly permuting the labels) was always positive and ranged from 0.121 to 0.507. In all cases, Rand was significantly different from zero indicating that the networks within a dataset were more like each other than randomly shuffled networks [48]. Additionally, the number of hub taxa was significantly different between treatment groups. However, the variation across datasets made the networks largely incomparable and no consistent patterns emerged in neither standard network metrics when comparing across networks (Appendix A) nor upon visual inspection of associations among target genera. Detailed descriptions of the network results for each of the individual studies can be found associated with Appendix A.

## 4. Discussion

We analyzed the raw data from three gut microbiota datasets of endurance athletes in search of bacterial species or genera consistently associated with extensive, intensive physical exertion or training. By controlling for the bioinformatics workflow and downstream statistical analyses across datasets, we were able to determine if there is a universal endurance microbiome. Overall, we did not detect any hallmarks of a subset of the gut microbiota universally associated with endurance events among these datasets. However there were limitations of the approach (see below). Similar conclusions have recently been reached in a literature review by Sato and Suzuki [46] (p. 4) who summarized their findings with, “Collectively, our findings suggest that intestinal microbiota diversity is more likely to vary among individuals than to be affected by ultramarathon.” We interpret our diversity, relative abundance and interactions results considering the original studies and the broader literature.

### 4.1. Diversity

Alpha diversity can be measured several ways (richness, Simpson’s index, Shannon’s index, etc.). In our analyses, there were no consistent differences in alpha diversity among the four treatment group comparisons. In comparison to the results previously reported from these datasets, Scheiman et al. did not report any diversity statistics [26], Zhao et al. [23] reported no significant differences in alpha diversity after finishing the half marathon, but did detect a change in diversity using taxonomic profiling [23], and Petersen et al. [18] detected higher diversity in a subset of their data (“Cluster Three”), but not a universal change in bacterial diversity associated with increased training duration [18]. More generally, several other studies have reported increased microbial diversity after an endurance event [6,16,19,20,51,52,53] (only when using Simpson’s index for cross country skiers in [20], though). However, there are also numerous investigations that found no change in microbial diversity in endurance athletes [10,20,22,54,55] (only when using Simpson’s index for marathon runners in [20]). Our results and the mixed results from the literature cited above clearly indicate that endurance training does not universally (nor even consistently) increase measures of bacterial diversity in the gut microbiota of endurance athletes.

### 4.2. Differences in Relative Abundance

In our analyses, none of the 16 target genera exhibited consistent and significant changes in relative abundance between treatment groups across all four comparison groups. The most encouraging results were from two genera that were significantly different in two of the four treatment group comparisons (*Romboutsia* and *Veillonella*). Nine of our sixteen target genera (56%) were significant in only one of the four treatment group comparisons and seven target genera (43%) were undetected in one or more of the datasets. In fact, one target genus (*Ezakiella*) was not detected in any of the three datasets. (These do not add to 16 since some genera were missing in one dataset and significant in another dataset, such as *Akkermansia.*)

The genera identified in two treatment group comparisons were from Boston marathoners “Athletes After” vs. Control and Chongqing half marathoners (*Veillonella*) and Chongqing half marathoners and competitive cyclists (*Romboutsia*), but both findings were curious. First, although we detected significantly higher abundance of *Veillonella* in “Athletes After” the Boston Marathon vs. sedentary Controls, Scheiman et al. reported *Veillonella* as the sole bacterial taxon differentiating “athletes before” vs. “athletes after”. (It was not significant in their “athletes after” vs. controls, but they did note that it was “more prevalent among runners than non-runners”). In both our analysis (Figure 3) and the original report [26] (their Figure 1B,C) *Veillonella* abundance was highly variable among individuals in the “athletes after” treatment group, even though our non-parametric statistical approach (Wilcoxon Rank Sum Test) should have accommodated these outliers (as was implemented in [26]). Unlike Scheiman et al., we averaged all samples from “athletes before” and “athletes after”. In contrast, Scheiman et al. performed regression against time before and after the marathon—suggesting pre-event host behaviors may have stimulated microbial growth (see Figure 1C in [26]). A separate study specifically investigated changes in *Veillonella atypica* using qPCR following six weeks of “endurance training” and found no significant differences between athletes vs. controls, which highlights the variable results among endurance studies [56].

For the second genus detected in two datasets, Zhao et al. [23] discuss *Romboutsia* in their text, but in their main figure, *Romboutsia* appears to have been replaced with “*Phaseolus sativus*”, a flowering plant in the pea family. Assuming this legume is truly *Romboutsia* in disguise, both our analysis and that of Zhao et al. show it decreases in relative abundance after the half marathon. However, *Romboutsia* abundance increased in the high-training group of Petersen et al. (2017) and in our reanalysis thereof. Unless the effects of *Romboutsia* are opposite in cyclists vs. half-marathoners, it appears that even the strongest candidates for endurance-associated species are inconsistent at best.

Outside of these four treatment group comparisons, several studies have compared relative abundance of bacterial taxa potentially associated with endurance (recently reviewed in [57,58]). Each study reports a different number and identity of the bacterial genera with differential abundance [10,16,19,20,22,46,51,54,59,60,61,62,63,64]; (also see Appendix A). Miranda-Comas’ review of athletes’ gut microbiota reports, “Although there is great variation in studies, *Faecalibacterium prausnitzii*, *Roseburia hominis*, *Akkermansia muciniphila*, and *Prevotella* species are some of the most commonly referenced as healthy or health-promoting gut species” [65] (p. 3). The inconsistencies in the number and identity of the bacterial genera associated with endurance from a wealth of independent studies question the existence of universal components.

### 4.3. Changes in Associations

Potentially, the universality of endurance-associated microbiota is not based on changes in abundances, but instead more subtle (yet consistent) changes in *associations* among bacterial taxa due to the unique gut environment of endurance athletes. However, after examining all pairwise correlations and network connectivity analyses, we conclude there are no consistent constitutive microbial associations, nor any induced by endurance events (except the previously described negative correlation between *Prevotella* and *Bacteroides*). We must be cautious when interpreting these association and network results as not all interactions represent true ecological relationships [41,66].

These largely inconsistent results among datasets could be caused by different starting points among individuals’ gut microbiota. For example, we wondered “Are all bacteria present in all individuals, just waiting for the right environment to grow, or is the microbiota constrained by bacterial colonization?” Restated, “Are endurance-associated microbiota growth-limited or colonization-limited?” Addressing this requires better understanding the sensitivity and limits of detection of current approaches to microbiota analysis. Scheiman et al. [26] (p. 1109) blur the boundaries between the two hypotheses in writing that “An important question is how this performance-facilitating organism first came to be more prevalent among athletes. We propose that the high-lactate environment of the athlete provides a selective advantage for colonization by lactate-metabolizing organisms such as *Veillonella*.” Some degree of colonization limitation could logically encourage the use of probiotics (e.g., “Nella”). Our lack of consistent correlations among datasets using non-parametric Spearman rank correlations and the network approaches, and repeated references to the large inter-individual variation in gut microbiota throughout the literature, also suggests that gut microbiota of endurance athletes are largely colonization-limited, and that most bacteria are not omnipresent in the gut awaiting the right environmental cues. Whether introducing target taxa such as *Veillonella* or *Romboutsia* into endurance athletes’ gut microbiota will influence their metabolism will require much broader sampling (but see Scheiman et al., 2019 additional evidence in germ-free mice).

One of the few consistent results we found in searching for bacterial associations was the excess of positive correlations compared to negative correlations. This was detected in all datasets yet is likely a technical limitation—and more evidence that our methodology is sound [67,68]. Although one might be tempted to infer a higher frequency of commensal relationships (producing positive associations) compared to competitive interactions (producing negative associations), Badri et al. explains that “the positive skewness may also be due to technical limitations in the data generation process and shortcomings in current statistical estimation. For instance, truncation to zero effects for low sequencing read counts likely obstructs unbiased estimation of negative correlations” [39] (p. 11). 

### 4.4. Limitations

Early gut microbiota studies classified most samples into one of a few narrow enterotypes, especially with regard to the ratio of Firmicutes to Bacteroidetes. However, thorough meta-analyses clearly showed that these enterotypes were not reflective of the true complexities of the microbiota [28,69]. Our reanalysis of existing datasets similarly suggests that there is no unique association of endurance-related gut microbes shared by all endurance athletes, at least as measured by 16S rRNA gene amplicon sequencing.

Our results have several limitations, some of which are technical, and others biological. One disconcerting practical limitation was that many of the published datasets we sought to reanalyze did not have publicly available 16S rRNA gene sequence data for reanalysis (Appendix A). For datasets we could access, we sought to compare similar studies, but this was challenging. For example, Scheiman et al. [26] and Petersen et al. [18] use 150 bp reads and Zhao et al. [23] used 250 bp reads. In hopes that any universal patterns describing endurance-associated microbiota would be agnostic to the length of the reads and the primers used for amplification thereof, we chose to compare these datasets (however, see the discussion below on 16S primer-bias that may have limited our ability to detect universal endurance-associated microbiota in comparing these studies). Additional concerns not addressed here are methods of fecal collection (time of day, type of tube, and storage temperature), some of which may affect the relative abundances estimated from 16S rRNA amplicon sequencing [31,70].

Another technical limitation surrounds the use of putatively “universal” primers for the 16S rRNA gene. Several authors have pointed out that, although the use of 16S data for determining microbial composition of the gut is powerful [71], the standard primers may not amplify all bacteria in equal proportions to their starting population sizes [31,72]. However, the three studies and datasets that we reanalyzed used the same or similar primers targeting the v4 region, so any PCR bias should have affected all datasets similarly. This issue is, therefore, unlikely to explain our failure to detect a universal endurance-associated microbiota. However, because we narrowed our focus to genera for which more than 75% of samples had reads mapped, it is possible that microbes whose rRNA sequences are poorly amplified by these universal primers could have been neglected.

A more biological limitation of our study is the limited taxonomic granularity we were able to apply. We narrowed our master taxa list to include only taxa with genus-level taxon identifications based on our 95% threshold [35,36,37,38,39], but it is well known that different microbial species within a genus may have distinct impacts on human metabolism and physiology (e.g., *Clostridium*) and may be responding differently to endurance. We may therefore have missed species-level or even within species (strain-level) changes or interactions relevant or unique to endurance athletes. More broadly, a 16S rRNA profiling approach will be largely blind to changes in microbial populations related to mobile genetic elements such as plasmids, which can significantly impact microbial physiology [73].

Functional redundancy—in this case, distinct microbial species capable of the same metabolic functions—could explain a lack of consistent endurance-associated species in these datasets. Moya and Ferrera (2016) invoked functional redundancy to potentially explain both the diversity of microbiota across individuals, and the relative stability of microbial communities within individuals to perturbation. Applying this principle to endurance-associated microbiota, although *Veilonella atypica* was identified as being enriched in a sample of Boston marathoners post-race, it could plausibly be substituted in diverse human samples by other gut microbes also expressing a pathway for converting lactate to propionate. Indeed, various authors (e.g., Carey and Montag, 2021) have proposed that microbial production of short-chain fatty acids such as propionate could be a key to microbiota-associated performance improvements in athletes. It would be useful to extend compositional studies of gut microbiota through metagenomic tools such as PICRUSt2 (Douglas et al., 2020) to look for other metabolic pathways shared in microbes potentially responsive to endurance training or activities.

Generally, because there are so many factors that can affect the gut microbiota, any comparison must be sufficiently controlled and have adequate sample size to account for these factors. Diet, which can be highly variable and is often not well-controlled in human research studies, is undoubtedly one of the largest factors affecting the gut microbiota. Challenges related to diet were clearly evident in the studies we reanalyzed. The study of Chinese half-marathoners examined here showed a response in the microbiota in just one day to the chemistry of the dietary intake. Clarke et al. found that athletes had higher protein intake, which may have affected microbial diversity in athletes’ gut microbiota, regardless of their endurance-induced physiological differences [16]. Indeed, life-long effects of early diet on the gut microbiota have been repeatedly reported [28,74], that could constrain future exercise-induced changes. In addition to diet, there are dozens of other environmental factors that strongly impact the composition of the gut microbiota. A rule of thumb for a logistic regression is 10–20 samples per parameter being estimated [75] which suggests on the order of 100–200 samples to account for the numerous covariates affecting the gut microbiota. The number of covariates can be reduced (but not eliminated) by using a paired sampling scheme (i.e., the same athlete before vs. after); however, this approach fails to detect long-term changes in gut microbiota that differentiate endurance athletes from sedentary controls.

Finally, we conducted a power analysis to estimate the sample size necessary to detect significantly different abundances between treatment groups. We used the empirically determined “d” based on the ability to detect a 10% difference in the average abundance per genus and calculated the pooled standard deviation among all samples per genus. We applied a power of 0.8 (an 80% chance of concluding there’s a real effect) and a Bonferroni-corrected alpha based on the number of genera detected per dataset assuming that many tests will be conducted. Our power analysis suggests that all four treatment group comparisons require substantially larger sample sizes than those in the published studies, ranging from ~150-fold more samples (Chongqing half marathon) to ~800-fold more samples being necessary (Boston marathon Athletes After vs. Controls) (Appendix A). Even using the largest “d” value among all genera detected per study, the sample sizes in these three studies are still 10-fold to 150-fold smaller than necessary. Only after thoroughly controlled studies are conducted on substantially larger sample sizes will investigators be able to reliably assess whether intensive exercise affects the gut microbiota amidst a background of covariates.

### 4.5. Future Research

Rather than “universal” endurance-associated microbiota, it must be considered that different types of athletic activity may place different demands on the anatomy and physiology of participants, which may differentially influence the gut microbiota. Tabone et al. write that “Exercise frequency, intensity, performing time, type of exercise, exercise volume and progression are all factors that influence physiological responses and exercise adaptations, and will need to be considered in future studies investigating the beneficial effect of exercise on the gut microbiota” [22] (p. 8). Due to inter-individual variation, improved control could be gained by comparing the same individual sampled repeatedly before and after endurance events, then pooling across studies. Sedentary control groups could be thoroughly sampled, under diet-controlled conditions, then trained to become an endurance population and sampled during the process. Pugh et al. recently reviewed the differences between the gastrointestinal health of female endurance athletes remarking, “The links between female microbiome, estrogen, and systemic physiological and biological processes are yet to be fully elucidated” and that “Many of the male-female differences seen (e.g., in immune function) may be, at least in part, influenced by such GI related differences” [76] (p. 755). Given these complexities, our search for the existence of a universal endurance-associated gut microbiota may be destined to mirror the mixed results of probiotic use in sports [13]—elusive at best, and potentially not a universal phenomenon.

## Figures and Tables

**Figure 1 microorganisms-10-02213-f001:**
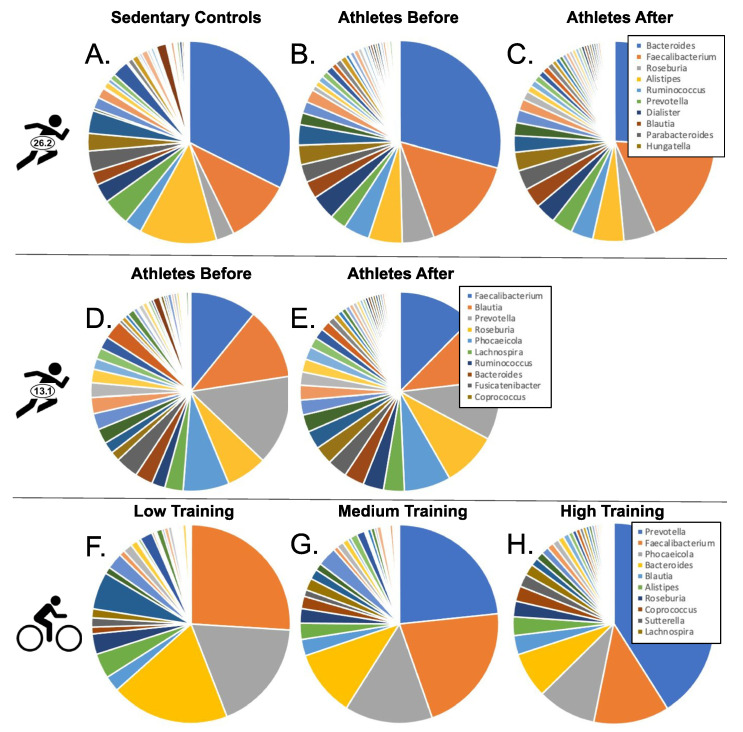
Average proportions of bacterial genera in the eight comparison groups. Data are from Boston Marathoners (**A**–**C**); Chongqing Half-Marathoners (**D**,**E**); competitive cyclists (**F**–**H**). The top 10 most abundant genera for the highest performing treatment group are indicated in the legend for each dataset.

**Figure 2 microorganisms-10-02213-f002:**
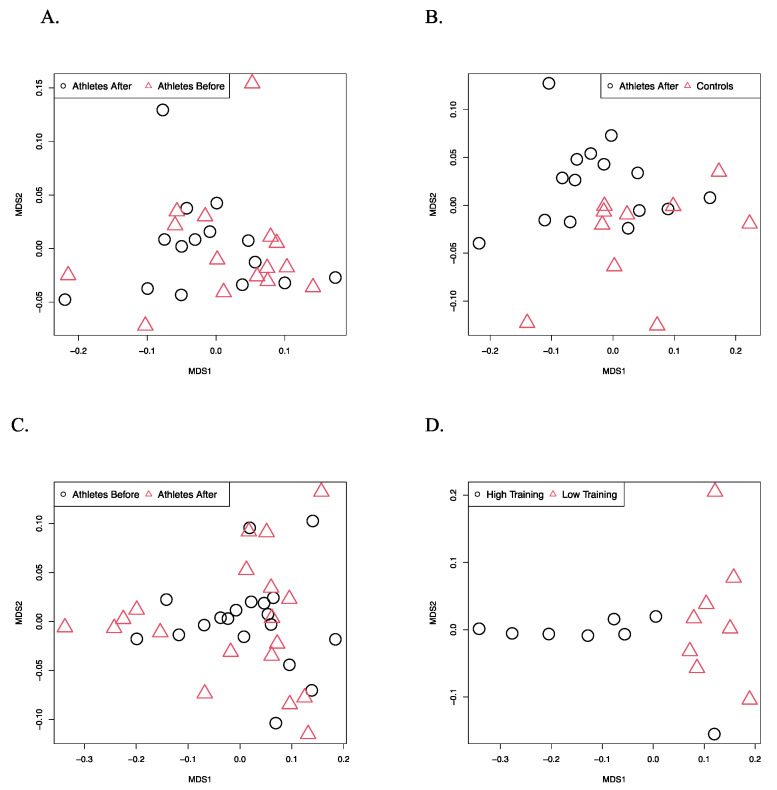
MDS plots of ANOSIM distances for the four datasets. (**A**) = Athletes Before vs. Athletes After Boston Marathon. (**B**) = Controls vs. Athletes After Boston Marathon. (**C**) = Athletes Before vs. Athletes After Half Marathon. (**D**) = Cyclists Low- vs. High-Training Groups.

**Figure 3 microorganisms-10-02213-f003:**
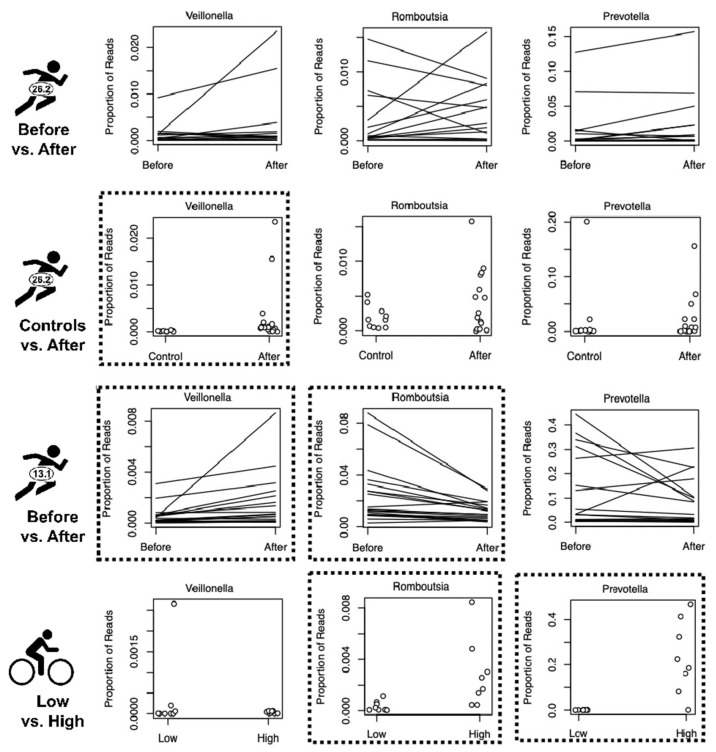
Relative abundance for three selected bacterial genera comparing Boston Marathoners before vs. after (1st row), sedentary controls vs. athletes after (2nd row), Chongqing Half-Marathoners before vs. after (3rd row), and competitive cyclists in the low-volume vs. high-volume-training groups (4th row). Lines connect paired samples of the same individual (before vs. after), whereas points compare samples taken from different individuals (controls vs. athletes or low- vs. high-training groups). Significant differences between treatment groups are indicated with boxes with dotted borders.

**Figure 4 microorganisms-10-02213-f004:**
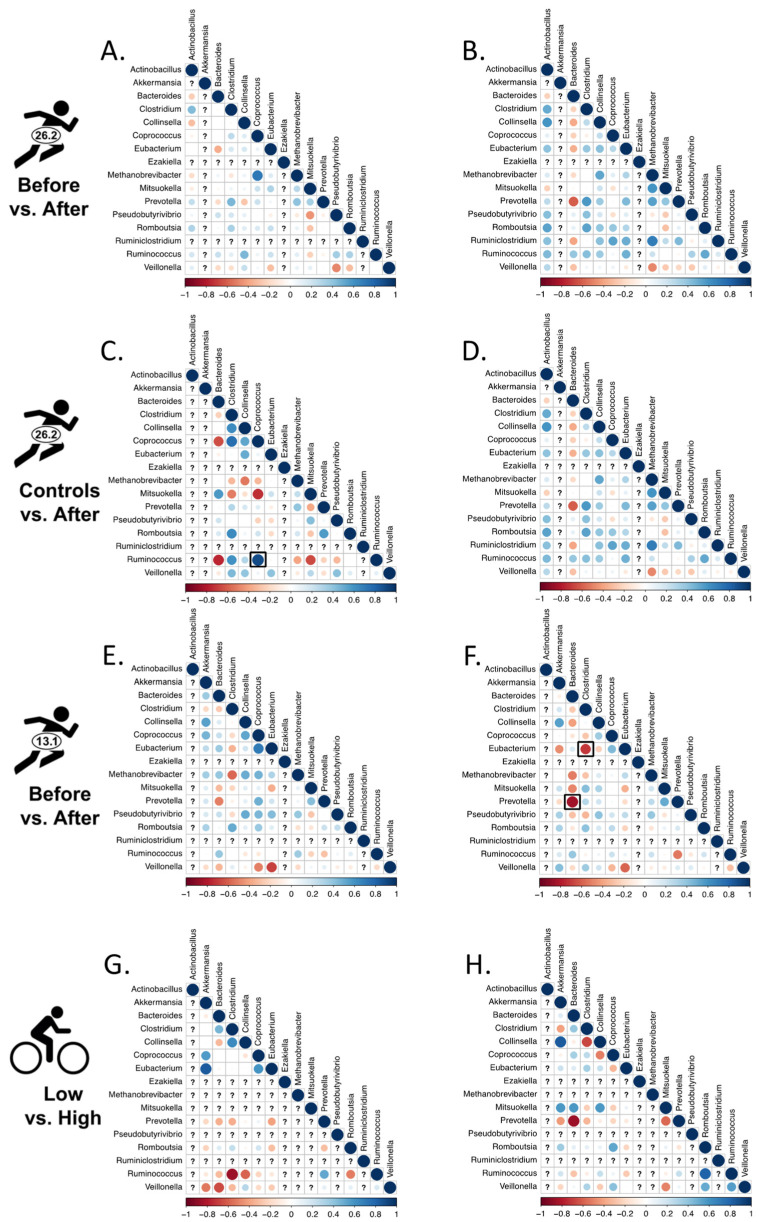
Pairwise Spearman’s rank correlations for target genera. The direction of the correlation (r) is indicated with the color of the circle (red = negative correlation; blue = positive correlation). Stronger correlations are indicated with greater color intensity and larger diameter of the circle. Significant correlations are boxed in black (*p* < 0.05 after Benjamini–Hochberg correction). Question marks represent taxa that were not detected in each dataset. Athletes before (**A**) and after (**B**) the Boston Marathon; controls (**C**) versus athletes after (**D**) the Boston Marathon; before (**E**) and after (**F**) the Chongqing Half Marathon; high (**G**) and low (**H**) training groups of USA competitive cyclists.

**Table 1 microorganisms-10-02213-t001:** Datasets used in this work and number of microbial genera detected in our bioinformatics pipeline.

Event and Location	Treatment Group	Sample Size	Sampling Frequency	Reference	No. of Genera Detected
Boston Marathon, Boston, MA, USA	Runners Before	15	Multiple samples taken before the event	[26]	221
Boston Marathon, Boston, MA, USA	Runners After	15 (paired with above)	Multiple samples taken after the event	[26]	233
Boston Marathon, Boston, MA, USA	Sedentary Controls	10	Multiple samples taken from controls	[26]	228
Chongqing International Half Marathon, Chongqing, China	Runners Before	20 runners ^1^	Once before the event	[23]	194
Chongqing International Half Marathon, Chongqing, China	Runners After	20 (paired with above)	Once after the event	[23]	197
Competitive Cyclists, USA ^2^	Low (6–10 h/wk)	8	One time point	[18]	115
Competitive Cyclists, USA ^2^	Medium (11–15 h/wk)	17	One time point	[18]	133
Competitive Cyclists, USA ^2^	High (16–20+ h/wk)	8	One time point	[18]	115

^1^ One before-event sample (BEF09) could not be used because it had an inconsistently formatted fastq file (Zhao attempted personal communication, unrequited). ^2^ Petersen et al. [18] reports very high consistency in relative abundances estimated from both 16S amplicon sequencing and whole-genome shotgun sequencing. We reanalyzed the 16S results for direct comparison with other endurance datasets.

**Table 2 microorganisms-10-02213-t002:** ANOSIM results comparing overall differences between treatment groups for four datasets.

Event	Treatment Group	Number of Individuals	Number of Bacterial Genera	R-Value	*p*-Value	MDS Stress
Boston Marathon	Athletes Before vs. Athletes After	15 (paired)	282	−0.021	0.6556	0.093
Boston Marathon	Controls vs. Athletes After	10 vs. 15	282	0.093	0.0998	0.074
Half Marathon	Athletes Before vs. Athletes After	19 (paired)	198	0.002	0.3603	0.104
Professional Cyclists	Low- vs. High-Intensity-Training Group	8 vs. 8	148	0.517	0.0012	0.060

**Table 3 microorganisms-10-02213-t003:** Relative abundance statistical results. *p*-values from Wilcoxon rank sum tests with continuity correction for the relative abundance comparisons among 16 target genera across the four treatment group comparisons. No correction for multiple testing was applied to the target genera since these target genera were predicted to be significant a priori. A plus sign (+) following the *p*-values indicates that the abundance in the endurance treatment group was greater than the control group. A minus sign (−) indicates that the abundance in the endurance group was less than the control group.

Genera	Boston Marathon		Chongqing Half Marathon	USA Competitive Cyclists ^3^
	“Athletes Before” vs. “Athletes After”	“Athletes After” vs. Controls		
*Actinobacillus*	NS ^1^	NS	NA	NA
*Akkermansia*	NA ^2^	NA	0.038 (+)	NS
*Bacteroides*	0.07	0.09	NS	0.0070 (−)
*Clostridium*	0.04 (+)	NS	NS	0.065
*Collinsella*	NS	NS	0.0005 (+)	NS
*Coprococcus*	NS	NS	0.0003 (+)	NS
*Eubacterium*	0.07	NS	0.023 (+)	NS
*Ezakiella*	NA	NA	NA	NA
*Methanobrevibacter*	NS	NS	0.09	NA
*Mitsuokella*	NS	NS	0.0046 (+)	NS
*Prevotella*	NS	NS	0.07	0.00031 (+)
*Pseudobutyrivibrio*	NS	NS	NS	NA
*Romboutsia*	NS	NS	0.0002 (−)	0.0047 (+)
*Ruminiclostridium*	NS	NS	NA	NA
*Ruminococcus*	NS	NS	0.001 (+)	NS
*Veillonella*	NS	0.0019 (+)	0.0004 (+)	NS

^1^ NS indicates the relative abundance was not significantly different at alpha < 0.05); ^2^ NA indicates this genus was not detected in our analysis; ^3^ Low- vs. High-training group comparison.

## Data Availability

All read counts and proportions from our Geneious workflow are available from Figshare (DOI: 10.6084/m9.figshare.c.6036347). We have also included all the R scripts used for statistical analyses and our Geneious workflow that can be downloaded and run in Geneious Prime 2021.2.2 under the same Fishare DOI.

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
