# Peer review of "Is There a Universal Endurance Microbiota?"

_microorganisms, 2022, doi:10.3390/microorganisms10112213_

Round 1

Reviewer 1 Report

The manuscript no. 1958588 entitled “Toward a Universal Endurance Microbiome” by Olbricht et al. investigated, in silico the “endurance” microbiota. To my understanding, the study aimed to identify the core microbial signature in athletes or after endurance compared to sedentary control or status. The authors retrieved the 16S V3-V4 microbiota data available online from four different studies and reanalyzed datasets together using a newly created bioinformatics pipeline. While the topic is interesting, this paper presents several issues, and revisions are required before granting acceptance. I have major concerns regarding the data analysis and methodology that the authors should give serious consideration.

The authors are refereeing to microbiome data in their analysis, while by definition, this should be renamed microbiota, the authors are specifically talking about microbial composition; there is no genomic or metagenomic analysis. The term microbiota should be more appropriate in this manuscript.

Olbricht and colleagues pooled the V3-V4 16S data to be reanalyzed. It is known that microbiota data are very variable and results are not consistent from one study to another. Moreover, sample collection, storage, DNA extractions, PCR, and sequencing, are all sources of variations; it is not clear how the pooled metadata was normalized and if all these variables/co-variables were taken into consideration. The authors may need to reanalyze the data differently and use a more sophisticated pipeline to be able to identify the core microbial changes in the gut and markers of endurance or sedentary. The data as presented make it difficult to draw a conclusion.

In addition to the 16S data, the authors could also infer metabolic pathways (for example using PICRUST) to be able to identify key potential pathways changing in response to exercise.

Intense physical activities are known to be associated with the production of SCFA, however, there is no mention of SCFA-producing bacterial taxa in the text.

Reviewer 2 Report

Olbricht et al. present a manuscript describing a meta-analytic study of gut microbiome associations to endurance-based exercise in physically active and presumably healthy adults. The study's primary objective is to describe additional details of microbial taxonomy within the endurance enteric microbiome via uniform bioinformatic processing and statistical analysis of pre-existing data. More specifically, the study aims to identify core microbiome elements that are consistent throughout the source material. The main conclusion drawn from the analysis was that taxonomic measures are dissimilar between the datasets. This conclusion aligns with previous observations, however its significance is obscured by limitations in the study design, methodology, evaluation of results, and investigative intention.

Accordingly, a number of major points require addressing.

The statistical approach contradicts best practices (violating test assumptions) and itself (using Pearson for liberal correlations vs SparCC to reduce false positives). It's highly likely (as the authors point out) that the data are non-normally distributed and, accordingly, non-parametric tests should be used. Similarly, despite acknowledging the compositional nature of the data (Gloor et al. 2017), and treating a portion of the data as such ( clr transformation for network construction), it's unclear why these practices where not applied elsewhere. The R release and all R libraries (e.g., vegan) should have corresponding version numbers. Given the overall aim of the study, it was surprising to see a lack of statistical assessment of similarity between the cohorts (e.g., ANOSIM; vegan::anosim). Following from this, an ordination approach would be a valuable graphical strategy (e.g., PCA bi-plot, nMDS, etc.).

More fundamentally problematic is the collection of samples used for the basis of the study. Despite the many obvious confounds (age, gender, diet, etc) the nature of the samples may be misused in the present approach. An interesting and potentially important point that the authors touch on is the dichotomy distinguishing the adaptive community driving fitness and the responsive community (ie recovery). Regrouping the samples from all the studies, for example, as pre-event (nourished and primed for energy use) and  post-event (depleted and primed for recovery) may better identify cross-study trends. That said, taxonomy may be an inappropriate measure to pursue in general given that it's the functionality of microbes that serves as the basis of their ecological placement (e.g., mucin degradation, carbohydrate conversion, etc). As such, metabolic redundancy may obscure the shared importance/role of different taxa. At a minimum this point should be addressed in the discussion.

The authors may benefit from reevaluating the aim of the study as it becomes unclear if the purpose is to demonstrate that variation in bioinformatic pre-processing and statistical analysis modifies the outcome or if the focus is indeed an examination of the variable response to exercise, which as the authors highlight is underpowered to do so.

Additionally, there are smaller issues related to writing:

page 2 'more than 100 factors', probably but where is the value from? 'carbo-loading', typo? Do bacterial associations necessarily have to be beneficial?

page 3 'the after the race', typo.

page 4 BLAST identifying diversity? Is this meant to refer to taxonomic assignment? May be worth stating that OTU clustering was used to construct reduced DBs, also reference to 16s database is missing

page 5 confirm DOI is correct (seemed to direct incorrectly). 'These genera...' awkward sentence.

page13 Fig 3 why include taxa that are completely absent?

page15 '...consistently significant different abundances...' awkward wording. Should this be 'encouraging'?

page16 'Unless the effects...' again, it may not be meaningful to compare these cohorts. 'The negative correlation between these two cosmopolitan genera...' Entire point is lost. 

page17 Entire first paragraph is completely unclear and seems unnecessary... as is second paragraph

'...clearly indicates that there is no universal gut microbial community shared by all endurance athletes...' Completely unsubstantiated claim. Similarly for final sentence on page 19.

Round 2

Reviewer 2 Report

The authors have graciously considered the majority of my comments. In the current version of the manuscript there are a few points to be addressed.

Page 2: 'In fact, more than 100 factors...'. The reference appears to be unchanged, and Cerda et al. does not report on >100 composition influencing factors. Rephrasing to something along the lines of 'many factors' may be more appropriate without a source that offers a quantification in this regard.

Page 15: Table 3. The meaning of signed values (+/-) is not clearly stated.

Page 20: 'In contrast Scheiman et al. performed...'. Indeed, the suggestion that 'bacteria can anticipate the upcoming endurance event' is absurd. However, it does not seem that this was asserted by Scheiman et al., and rather the generalized linear mixed-effect modelling indicated that pre-event host behaviours stimulated microbial growth. Adjusting the statement to reflect a more sensible original interpretation of the work would be beneficial.
